# Using Functional Annotations to Study Pairwise Interactions in Urinary Tract Infection Communities

**DOI:** 10.3390/genes12081221

**Published:** 2021-08-06

**Authors:** Elena G. Lara, Isabelle van der Windt, Douwe Molenaar, Marjon G. J. de Vos, Chrats Melkonian

**Affiliations:** 1Systems Biology Lab, AIMMS, Vrije Universiteit, 1081 HZ Amsterdam, The Netherlands; elena.garcia.lara@hotmail.com (E.G.L.); d.molenaar@vu.nl (D.M.); 2GELIFES, Universtity of Groningen, 9747 AG Groningen, The Netherlands; isabelle.vanderwindt@hvhl.nl

**Keywords:** urinary tract infection, microbial community, microbial interaction, gene annotation

## Abstract

The behaviour of microbial communities depends on environmental factors and on the interactions of the community members. This is also the case for urinary tract infection (UTI) microbial communities. Here, we devise a computational approach that uses indices of complementarity and competition based on metabolic gene annotation to rapidly predict putative interactions between pair of organisms with the aim to explain pairwise growth effects. We apply our method to 66 genomes selected from online databases, which belong to 6 genera representing members of UTI communities. This resulted in a selection of metabolic pathways with high correlation for each pairwise combination between a complementarity index and the experimentally derived growth data. Our results indicated that *Enteroccus* spp. were most complemented in its metabolism by the other members of the UTI community. This suggests that the growth of *Enteroccus* spp. can potentially be enhanced by complementary metabolites produced by other community members. We tested a few putative predicted interactions by experimental supplementation of the relevant predicted metabolites. As predicted by our method, folic acid supplementation led to the increase in the population density of UTI *Enterococcus* isolates. Overall, we believe our method is a rapid initial in silico screening for the prediction of metabolic interactions in microbial communities.

## 1. Introduction

The microbiome of our body has a significant impact on our health and disease, as disequilibrium in these microbial communities may lead to dysbiosis [1]. The interactions between the microbes in a community may come in many forms and can have different effects, including synergy or inhibition of the growth of other community members, killing by toxins, or cooperative behaviour induced by quorum sensing [2] or the exchange of nutrients [3,4]. Moreover, interactions between the members of a microbial community can potentially alter the sensitivity to antibiotics in vitro, and, therefore, potentially the effect of treatments [5].

Several tools from biology systems have been used to study microbial interactions, with a focus on the analysis of the flow of molecules and energy through metabolic networks [6]. Genome-scale stoichiometric models have been used to study single organisms and simple communities to high levels of detail [7,8]. However, model reconstruction and manual curation are still a time-consuming process. Simpler and more coarse-grained models were proven to be useful, as they have less accumulated biases and are more readily to be applied [8,9,10]. Both approaches rely on functional annotations predicted from genomic data. These annotation elements (e.g., from KEGG [11], MetaCyc [12], TheSEED [13], or EggNOG [14] databases) are a proxy of the putative metabolic capacities of an organism. Using these annotations, one could construct the topology of a metabolic network with the aim to infer the biochemical environment of an organism [15]. Subsequently, graph theory can be used to compare the topology of metabolic networks between pairs of organisms to compute interaction indexes, such as biosynthetic support score (NetCooperate [16]), complementarity or competition (RevEcoR [17]). Alternatively, it is possible to unveil properties of the system without inferring the whole metabolic network by comparing patterns in the KEGG Ortholog (KO) content of the organisms within a community [18].

One hallmark of urinary tract infections (UTI) in elderly patients is the presence of multiple pathogens [19]. In addition, although the urinary tract has historically been considered sterile in the absence of infection, it has recently been discovered that there is a resident microbiome [20,21,22,23]. Additionally, in uncomplicated UTIs there is often a background microbiome present, even though the contribution of the microbiome to the pathology is not yet clear [24,25]. Given that UTI affects 150 million people worldwide each year [26], and an estimated 50% of women report having had a UTI at some point in their lives [27,28], it is important to understand the potential role of bacterial interactions in these infections. In the UTI context, the main focus has historically been on *Escherichia coli* , the main urinary pathogen. Other taxa commonly involved in polymicrobial UTIs include *Klebsiella pneumoniae*, *Proteus mirabilis*, *Pseudomonas* spp. and *Staphylococcus* spp., and *Enterococcus* spp. [29,30,31].

The aim of this work was to study the genetic basis of pairwise interactions of the bacteria involved in polymicrobial UTI reported by de Vos et al. [5]. We used a pure data-driven approach to find putative metabolic interactions using genomic information and functional annotation of genes. Because the genome sequences of the organisms were not available, we collected 66 genomes from online databases to artificially constitute this UTI community. Then, we devised four complementarity and four competition indices for each pair of genomes based on the KOs for each KEGG pathway. Those indices were compared using the correlation with the growth variance from pairwise experiments [5]. The chosen complementarity measure was further used to select the most informative pathways, which may explain the positive growth synergies or lack of between pairs of strains. Overall, the results led to the hypothesis that members of a UTI community could provide intermediates of specific metabolic pathways to *Enterococcus* spp. In particular, the hypothesis that folic acid produced by the community members may promote the growth of enterococci was confirmed experimentally.

## 2. Materials and Methods

### 2.1. Yield and Growth Rate in UTI

The starting point of the project was data from pairwise combinations of bacteria from UTI communities. Communities containing four different species were isolated from hosts diagnosed with a polymicrobial UTI [19]. In an effort to analyse the interplay between these species, de Vos et al. [5] had quantified these interactions by means of phenotypic growth essays. Briefly, 72 bacteria strains were isolated from 23 UTI patients in a previous study ([19]), and cultured in a modified artificial urea medium (AUM) [32] (unconditioned medium, $N_{u}$) for 48 h. Afterwards, the supernatant from this cultures was used to supplement a second medium (conditioned medium, $N_{c}$) for 24 h. Interactions were calculated as the ratio of growth between conditioned and unconditioned media; positive interactions were due to an increased growth in conditioned medium, and negative interactions were due to decreased growth in conditioned medium, compared to the growth in unconditioned medium [5].

This analysis resulted in a pairwise interaction matrix G (Appendix A), of size *N* × *N*, where *N* equals to the number of strains isolated from UTI patients (72). The donors are represented in the y-axis and the acceptors in the x-axis. The strains are ordered by phylogeny; similar genera show similar behaviour [5]. The conditioned medium was partly mixed with AUM, so that the complemented medium ($N_{c}$) had between 0.6 and 1 times the original concentration of nutrients, depending on the consumption of the donor. Together, this approach allowed to distinguish between interactions with different signs (Appendix A). In the matrix, the strains are divided into six genera: Ecoli: *E. coli*; Ent: *Enterococcus* spp. (*E. faecalis* and *E. faecium*); KECS: *Klebsiella* spp. (*K. pneumoniae* and *K. oxytoca*), *Enterobacter cloacae*, *Citrobacter koseri*, *Serratia liquefaciens*, and *Pantoea sp4*; Pm: *P. mirabilis*; Mm: *Morganella morganii*; and Ps: *Pseudomonas* spp. (*P. aeruginosa* and *P. fluorescens*); St: *Staphylococcus* spp. (*S. aureus*, *S. haemolyticus* and *S. capitis*). Interestingly, the growth increase or decrease is conserved within genera [5], fencing the interactions to the core functions, and allowing us to create an approximate community to study.

### 2.2. Description of the Dataset

We mimicked the experimental UTI communities, using online available genomes, because the genomic sequence of the phenotypically measured strains are not available. This in silico community consisted of the same species in a similar proportion. The genome dataset used in this project was retrieved from the National Center for Biotechnology Information (NCBI) [33]. More specifically, the strains were retrieved from the Prokaryote Genome database, which contains the genomes of more than 400,000 organisms, 350,000 of which are bacteria (as of 20 July 2021). Only the sequences with level of assembly ‘complete’ were chosen. When a species presented more strains available than the number of species present in the experimental community, the strains related with UTI according to literature were preferred. Otherwise, human isolates were chosen and, as a last resort, isolates from other sources were selected. Appendix A describes the initial selection of species and the equivalent strains, of which the results are described below. Appendix A describes a selection of the same species, but this collection only contains urine or human derived isolates. The results of the pipeline on those two genome selections (described in Appendix A) were very similar (Appendix A, Appendix A). Because the genome sequences were not derived from the strains assessed for the phenotypic growth matrix, a one to one comparison was not possible. Thus, we grouped the information of the strains into groups related to their genera to compare the phenotypically measured UTI isolates and the downloaded genomes. Overall, the distribution of groups looks similar to the original UTI community (Appendix A). With this in silico community, the analysis was performed following the flowchart in Appendix A. The project was carried out for the most part in Python 2.7.12, using the Pandas 0.19.2 library [34], and R 3.2.3-4 (within the RStudio environment). The scripts were stored in a GitHub repository (https://github.com/egarcialara/UTI_bacteria_interactions accessed on 1 July 2021).

The ribosomal 16S gene sequences were available for each of the isolates in the experimental community. The 16S sequences were blasted against three different databases. One was GreenGenes [35], a global database for sequences of this kind. On the other hand, the global prokaryote database from NCBI [36] served as an alternative database. Furthermore, a local database containing the genomes of the strains already selected for the in silico analysis, together with other strains belonging to the same species, was generated.

### 2.3. Functional Annotation of the Genomes

On one side, the genomes were downloaded from the GenBank database [37] using the Batch Entrez system for a fast retrieval. The nucleotide sequences of the genomes were translated into amino acid sequences using Prodigal [38]. It was followed by a BlastKoala search [39], to find the KEGG Orthologs associated with each of the organism sequences. The parameters were kept as default, with the exception of ‘taxonomy group of genome’: Bacteria and ‘KEGG genes database’: genus_prokaryotes. We made use of another parallel annotation that consisted in sequences downloaded directly from the Integrated Microbial Genomes database (IMG) [40] (Appendix A). The final annotation for each strain had the form of a matrix with two columns: one containing the K accession numbers of all the KOs, and a second column with boolean values, 1 if the KO is present, 0 if not. Moreover, a binary matrix B, N×M, of *N* KOs and *M* bacteria strains was used from [18], to have a visual representation of the KO content. Each entry $b_{i,j}$ represents the presence of KO *i* in the strain *j* with a 1, or 0 otherwise.

### 2.4. Functional Annotation Division into Smaller Segments

The KO vectors were broken down, from one long list to smaller vectors that correspond to the KEGG pathways. There are more than 500 pathways listed in the KEGG database, of which 274 had at least one KO in one of the organisms. In order to link the KOs to their respective pathways, the representational state transfer (REST)-style KEGG Application Programming Interface (API) was used. The link form http://rest.kegg.jp/list/pathway (accessed on 1 July 2021) allows to retrieve the complete list of KEGG pathways, whereas the form http://rest.kegg.jp/link/kos/mapxxxxx (accessed on 1 July 2021) serves to associate the pathways and their KOs. The *’mapxxxxx’* represents a KEGG pathway identifier space. In a similar way, the KEGG database offers other entry points, such as BRITE Hierarchy and modules [39]. A segmentation of the initial vectors into modules can be completed similarly with http://rest.kegg.jp/list/module (accessed on 1 July 2021) (retrieve all modules in KEGG database) and http://rest.kegg.jp/link/kos/Mxxxxx (accessed on 1 July 2021) (associate modules with their KOs), where, in this case, ‘*Mxxxxx*’ would be the KEGG module identifier. The third kind of partition chosen was one level above pathways, based on BRITE Hierarchy (BRITE Hierarchy files > Genes and Proteins > Orthologs and modules > KEGG Orthology (KO)—all categories (groups)). At the end of the Results section, a comparison between the segmentation in modules, pathways, BRITE terms, and the complete set of KOs was performed by comparing their performance in finding a pattern.

### 2.5. Interaction Matrices

The combination of the boolean KOs vectors from the two organisms in a pair was completed in the form of asymmetric interaction matrices C, N×*N*. In them, *N* corresponds to the number of different strains in the community. The values $c_{d,a}$ are the resulting of the comparison of two corresponding KO vectors: of a donor *d* and an acceptor *a*. The interaction was interpreted as either complementarity or competition. All the values were normalised by the total number of KOs in the vector, to facilitate the comparison between pathways (not explicitly stated below). Additionally, in order to make indexes asymmetric, many of them were divided by the number of KOs in the acceptor (only if stated).

#### 2.5.1. Complementarity

The complementarity indices were calculated based on the premise that metabolic functions present in one genome and not in the other pair member, could lead to the donor to supply the lacking compounds to the acceptor. This would be related with synergy, and would be expected to have a positive effect in the growth on the acceptor. At first, any intermediate is equally likely to be ‘complemented’. There are different ways to compute complementarity:Complementarity 1: directly as the fraction of the KOs that are present in the donor set but not in the acceptor;Complementarity 2: as the fraction of the total amount of KOs that the acceptor could have when complemented (by the donor) divided by the acceptor’s own possibilities;Complementarity 3: as Complementarity 2, but without taking into consideration the shared KOs (similar to Hamming distance for boolean vectors, but divided by A instead of total number of dimensions to make it an asymmetric index);Complementarity 4: same as Complementary 3, but divided by the putative total functionality. In this way, it is equivalent to Hamming distance, but it is symmetric.

#### 2.5.2. Competition

The competition indices were envisioned as the similarity of the metabolic potential of the organisms in a pair. The more similar, the highest similarity value, and the most expected putative negative effect on growth. This corresponds with the experimental set-up in which the complemented medium in the pair was only partially restored when transferred from the acceptor to the donor. Thus, a similarity in the compounds consumed by the former, would limit the latter’s uptake. The indices considered are as follows:Competition 1 : directly the fraction of shared KOs;Competition 2: equivalent to competition_1 but compared to the acceptor function to make it asymmetric;Competition 3: equivalent to competition_2, but compared to the total possibilities;Competition 4: is just Pearson correlation (just for comparison).

#### 2.5.3. Selection of Interaction Indices

We selected the interaction indices that matched the growth variations best, based on the correlation. Firstly, we selected the 100 pathways with highest standard deviation with respect to the pathway coverage between genera, as considered putatively, at least for the start, the most informative. Secondly, Pearson correlation was calculated for the index of each of these pathways and growth—either yield or rate. The ‘goodness’ of an index was tested by checking the correlation for each pathway interaction matrix with the growth matrices. In order to compare two matrices, they were flattened into a 1D array. Finally, a *p*-value was computed by permuting randomly the growth values, as in the equation that follows:$$P_{perm}=\frac{\sum_{n}^{N}\left|correlation_{randomized}\right|>\left|correlation_{original}\right|}{N_{permutations}}$$
where *N* corresponded with 1000 permutations. The indices with largest correlations were used for further study.

#### 2.5.4. Selection of Informative Pathways

We sought to identify those pathways in which there were a clear separation between pairs with high complementarity and increase in growth (ϵ > 0.22, Appendix A) and other pairs. In order to filter down the most informative pathways, we ranked the pathways depending on their performance in a classification between those two groups. We chose an ensemble of simple models, as they usually yield a performance as good as very complex models [41]. For this case, we chose three distinct methods: Boruta feature selection, as used in [18] with similar data, support vector machine (SVM) recursive feature elimination (RFE) as it performs well in many types of data and holds no assumptions [42] and a statistical Mann–Whitney U test (MWU) with which the selection is independent of the model. Boruta [43] is an extension of the random forest algorithm, which uses this classification to iteratively discard features. A Z-score of an attribute can be calculated by dividing the average accuracy loss for all the trees of a random forest by its standard deviation. In order to set an statistical significance Boruta calculates shadow attributes for each feature. A paired *t*-test is then calculated to interpret if the Z-score of the real attribute is higher than that of the maximum Z-score among shadow attributes. We ranked the pathways by the computed feature importance itself. The SVM algorithm classifies the points by representing them into space and creating a decision boundary that separates the groups. The optimal decision boundary is the one that maximises the minimum distance from the separating decision boundary to the nearest points or support vectors. Each feature has a weight on the decision vector, and this coefficient was used to rank the pathways. When applied RFE for selection, the feature with smaller ranking criteria, based on this weight, is removed iteratively [42]. Lastly, a model-free statistic method was selected. The MWU test assumes as null hypothesis that the probability of picking a value from a group bigger than the other group is equal. It is equivalent to a non-parametric *t*-test. The result is a statistic computed for each pathway and an associated *p*-values. The result was a ranking of the importance of the pathways, based on the MWU statistic, SVM RFEs score and Boruta’s importance. The ensemble was performed by summing up the rank position of each pathway. A stacking of base learners was avoided due to the small size of the dataset.

### 2.6. Alternative Approaches

In the project we compared the indexes above with two previously published tools that compute indexes based on metabolic potential, and that could be compared with the ‘complementarity’ indices presented above. NetCooperate [16] incorporates a metabolic complementarity index (MCI) that shows a putative synergy between two organisms. More recently, RevEcoR [17] has come as a tool that calculates Complementarity as “the fraction of compounds in species A’s seed set appearing in the metabolic network, but not appearing in species B’s seed set”, as well as Competition: "the fraction of compounds in species A’s seed set that are also included in species B’s seed set". The seed set refers to the predicted minimum set of compounds required for the growth of an organism. These three indexes were computed using their respective R packages, and the resulting matrix was analysed similarly as the rest of the indices.

### 2.7. Checking Transporters Availability

The biological potential of interactions between the pairs was checked by the presence of membrane transporters in the annotations of the organisms. This presence was assessed using KEGG, specifically three pathways: ABC transporters, phosphotransferase system (PTS), and bacterial secretion system.

### 2.8. Supplementation Experiments

A 1:1000 dilution of overnight culture pre-grown in modified artificial urine medium of *Enterococcus faecium*, isolated from a polymicrobial UTI [19], was inoculated with in 5 mL artificial urine medium [5]. Cultures were grown with and without supplementation of 2 μM serine, 2 μM glutamate, 1 μM histidine, 0.7 μM vitamin B12 for 24 h, at 37 °C, shaking 200 rpm. The effect of folic acid supplementation on the growth yield was assessed on two *E.*
*faecium* isolates and three *E.*
*faecalis* isolates, isolated from a polymicrobial UTI [19]. The 1:100 dilutions of overnight cultures pre-grown in modified artificial urine medium were inoculated in 96-well plates, with wells containing 200 μL artificial urine medium [5]. Cultures were grown with and without supplementation of 30 μM and 300 μM folic acid for 24 h, at 37 °C, shaking 500 rpm. Additional negative controls were artificial urine medium without bacteria, artificial urine medium with supplements without bacteria. To assess the effect of the addition of the metabolic supplementation on the population size, the yield, the optical density of the triplicate cultures was measured at OD600 nm after 24 h. Significant differences in growth yield due to folic acid supplementation compared to the control without folic acid supplementation, were assessed with a *t*-test (*p* < 0.05), with Bonferroni-correction to adjust for multiple comparisons.

## 3. Results

### 3.1. Complementarity Indices Show the Greatest Correlation with the Pairwise Interaction Growth Yields

The starting point was finding representative genomes of the species present in the UTI communities, as observed in [5]. Blasting the 16S sequences from the UTI isolates against global databases did not yield specific matches, as this approach could not distinguish beyond the species level of taxonomy. Thus, the representative strains were manually picked from the NCBI database based on their origin of isolation, by an order of preference from urine or UTI origin, human or animal, shown in Appendix A. The 66 selected genomes were divided into six groups. Each group corresponded to the genera of the isolate, with the exception of *Klebsiella* spp., *E.*
*cloacae*, *C.*
*koseri*, *S.*
*liquefaciens*, and *Pantoea* spp. which were grouped altogether (KECS group). For all genomes, the open reading frames (ORFs) were predicted, and the corresponding amino acid sequences were annotated as KOs using BlastKoala [44]. With the resulting KO vectors of each strain, we built a binary matrix *B*. This representation helps visualising how the KO are conserved within genera (Figure 1A), i.e., the KOs that are shared between species of a genus. The average annotation ratio of KOs in comparison with the gene sequences was of 60.6% (st.dev. 9.1%), although it was different between the groups. This, together with the different absolute number of genes originally present in each genome, caused that the total number of functional annotations varied depending on the genera. The Ps and KECS group (Appendix A) had at least twice as many functional annotations when compared to enterococci and staphylococci (Figure 1B). The lower number of absolute number of genes and annotation ratio is evident in enterococci and staphylococci (Gram+) (Figure 1A,B).

From the KO vectors we created interaction matrices, *C*, based on complementarity or competition indices (Appendix A). The values for each cell ($c_{d,a}$) correspond to the calculated index between the donor species *d* and the acceptor strain *a*. The heatmaps in Figure 2 represent the different strains, grouped by genera, which adopt the role of donors and that of acceptors in the y- and x- axis, respectively. A darker shade of black in the cells is associated with a higher value of the complementarity index. The computed measure is able to group the strains by their genera. These matrices had the same dimensions as the relative growth matrix as constructed in the growth variance experiments from [5] (reconstructed in Appendix A), which allowed to calculate the correlation between the interaction measures and growth (Appendix A).

The measures show a general behaviour across the pathways in their correlation with growth. The complementarity measures had a positive correlation with growth yield. In particular, Complementarity 3 ((D∪A−D∩A)/A) showed the largest average correlation (ρ= 0.41) with growth yield. Thus, it was chosen for the subsequent analyses (Figure 1C). A similar measure, Complementarity 2 ((D∪A)/A), had a similar average correlation (ρ= 0.33). Additionally, the *p*-values obtained from a permutation test were lower in the cases where the correlation was higher. Competition measures, such as Competition 1 (D∩A) showed a negative correlation of up to −0.54. However the *p*-values was not low enough to indicate that the index was informative (Appendix A). Moreover, no pattern was seen when comparing the interaction indices, both complementation and competition, to growth rate (Appendix A). The values for correlation between the interaction indices and the growth rate variation were low, with a minimum value of −0.36 (Competition 2). Nonetheless, the *p*-values were too large to hold onto these indices.

### 3.2. Analysis of Genomes in Terms of KEGG Pathways

In our analysis we chose to use the KEGG pathway segmentation because it is the best curated, as well as an intuitive way to break down the entire KO vector. However, one could construct similar interaction matrices without splitting the KO vector. With this approach, the relationship between the interaction measure and growth data in the organisms are lower in comparison to segmenting the KO vector by KEGG pathways (KO vectors correlation: ρ 0.62, *p* < 0.001, KEGG pathway segmentation: 0.66 (*p* < 0.001)) (Figure 2).

Next, we sought to identify the pathways for which the relationship between the interaction indices and the growth yield was most prominent. Subsequently, we ranked the most informative pathways using an ensemble of Boruta feature selection, support vector machine and the Mann–Whitney U test (MWU). Two-classes were used as targets for each method, with one representing the positive growth effect of the pairwise interactions and the other the neutral or negative growth effect. The threshold was set as ϵ > 0.22 as mentioned in [5]. The final selection was voted by all three methods by summing the pathway ranks of their individual result. The top ranked pathways are reported in Table 1.

On the other hand, using modules as another available segmentation, also underperforms (e.g., Glutathione metabolism, Phenylalanine metabolism, ‘Tyrosine metabolism’ have values of correlation 0.65, 0.63, 0.61, respectively, (*p*-values $\leq 2.5\;\times \;10^{-4}$), whereas their best scoring module correlations (and *p*-values) are 0.42 (0.01), 0.34 (0.04), 0.31 (0.06)). In addition, the modules have a very small size: 78% of them are comprised of less than 10 KOs, while only 18% of the pathways have that few KOs. Therefore, we concluded that the pathway segmentation of the KO sets was the most informative choice in this context (Figure 2D).

### 3.3. Enterococci Are Metabolically Complemented by the UTI Community

The interaction indices differ by genus. In particular, *Enterococcus* as an acceptor has a unique behaviour. In the pairs in which *Enterococcus* acts as an acceptor, the value of the complementarity index is higher than for other genera of the UTI community, which corresponds closely with the measured positive interactions affecting growth yield [5]. We mapped the KOs into the KEGG pathway maps to study the different availability of orthologs in the six genera in more detail. In accordance with the highest complementarity for enterococci, the patterns of selected pathways (Table 1) showed that enterococci lacked the orthologs that were present in the other genera (Figure 3. For example, in histidine metabolism, the functions that lead to histidine synthesis are present in most of the strains of *E. coli* but in almost none of the enterococci (Figure 4). A selected number of possible exchanged compounds is described in Table 2, based on the interpretation of the top ranked pathways.

### 3.4. Folic Acid Promotes Growth of Enterococci in Artificial Urine Medium

To validate the predicted positive interaction with a genetic basis in the folate pathway, we performed supplementation experiments. We compared the growth of few *E.*
*faecium* and *E.*
*faecalis* isolates from a polymicrobial urinary tract infection [5,19], which was cultivated in artificial urine medium with and without folic acid supplementation (Methods). We found that the supplementation of folic acid increased the growth yield of several *E.*
*faecium* and *E.*
*faecalis* isolates (Figure 5). Supplementation of the artificial urine medium with serine, glutamate, histidine, vitamin B12, and a combination hereof did not lead to an increase in the growth yield of *E.*
*faecium*.

### 3.5. Alternative Approaches Fail to Discover Relevant Patterns

We further compared our approach to create informative interaction indices with two previous published tools: RevEcoR and NetCooperate. Both are based in constructing the ‘seed set’ from the KO lists of each strain, and then using the network topology to compute the Complementarity or Competition (RevEcoR [17]), or the Biosynthetic Support Score (NetCooperate [16]). Using these indices we did not find a pattern to be shared with the data of growth yield or growth rate that we could further explore (Appendix A).

## 4. Discussion

Here, we have developed a rapid method that helps finding relevant information on the pairwise interactions of UTI isolates without the need for highly curated models. Since the genome sequences of the UTI isolates investigated in [5] were unavailable, we used publicly available genomes of similar organisms. Therefore, we have to accept an increased bias towards strain specific interactions. To compensate for this potential bias, we focused our analyses on the interactions conserved between genera. Furthermore, interactions outside of the core metabolism and potentially encoded by the pan-genome, like toxin synthesis or quorum sensing, are more likely to be missed since these are known to vary more between strains. This allowed us to use the genomes of strains not strictly associated with UTI to generate hypotheses on interactions between members of UTI communities. We obtained similar results compared to genomes more strictly related to UTI, which can be explained by the method’s focus to identify core-genome interactions or indicate that the environmental interactions signal is weak. We used the functional annotations of the genomes and we achieved an average of 60% annotation across the UTI selected genomes, while 70% is considered sufficient to represent the repertoire of each cell [48,49]. Additionally, functional metabolic pathways depend on the encoded genes, as well as mRNA and protein fluctuations. Therefore, we believe the extension of our approach by incorporation of transcriptomics or proteomics will achieve higher accuracy to predict the putative interactions.

The use of KEGG annotation as a ‘set of KOs’ proved to be useful. We mapped the KOs into KEGG pathways in order to identify KOs forming a coherent metabolic path. For future extensions, graph theory algorithms, which take into account the topology in a metabolic network are promising approaches. Not surprisingly, they are already proposed and used in the analysis of communities [50,51]. Such tools combine the use of the metabolic network topology and KO annotations to create scores for competition and cooperation. Specifically, we tested RevEcoR and NetCooperate tools, which yielded a lower performance in our investigation. The choice of using KEGG pathways for the segmentation of the whole genome came after the evaluation of different gene classifications. For instance, the division of the genome KOs in pathways allowed the discovery of relations in the data with higher resolution than when taking the whole set. Using the BRITE hierarchy, which provides another level of functional organisation than pathways, yielded an intermediate performance. On the other hand, KEGG modules (the smallest segmentation) may not be reliable due to their limited size. This suggests that the granularity of the segmentation is important, especially when the whole metabolic network is used.

The yield and growth rate of isolates grown in replenished conditioned medium was compared to the reference in the unconditioned medium [5]. This allowed the interpretation of negative interactions as either competition due to resource overlap or inhibition. Furthermore, positive interactions were also observed and interpreted as cooperation [5]. The creation of measures was based on complementarity and competition. On one hand, we assumed the complementarity indices maximise the difference of KO content in the pairwise association and, therefore, the number of exchangeable compounds. As such, the community dependency from the environment is minimised, reflecting cooperation and likely also robustness. Cooperative, cross-feeding interactions are especially likely to play a role in poor resource environments, such as urine (or AUM). On the other hand, we assumed competition indices minimise the difference of KO content in the pairwise association and, therefore, display the similarity of required compounds for all members of a given community [52].

In our study, complementarity had a positive correlation with growth yield, whereas competitive interactions did not show any strong relation. Additionally, neither of the measures correlated with the growth rate data. The distance or similarity measures have a strong impact on the outcome of the relations between the variables and the response [2]. In [53], an extensive comparison was made of different measures for community composition that could be applied to microbial interactions. In the UTI [5], as well as in other communities [31,54], it is reported that negative interactions between organisms are over-represented.

In our case, the creation of a complementarity index was more straightforward than a competition index. For the former, we could consider the increased metabolic capabilities that arise from an interaction. The latter would require an understanding of which specific metabolic processes are actually taking place. Therefore, it was more difficult to capture competition with the use of genomic information alone. Investigation of direct inhibitory interactions, as opposed to negative interactions by competition, was out of our scope. The data suggest that such interactions may form a large part, here 23%, of the total interactions [5] (Appendix A). In our analysis, out of eight measures, Complementarity 3 was chosen, because it had shown the largest average correlation across the metabolic pathways. As expected, a higher complementarity was linked to an increase in growth yield in the acceptor strain. This observation coincided with the dichotomy of Gram+ (*Enterococcus* spp. and *Staphylococcus* spp. strains) and Gram- (the rest: *E. coli*, *K.*
*pneumoniae*, *Proteus* spp.) bacteria, where the former group experience more positive interactions in growth [5]. This group also had a smaller number of metabolic genes, accompanied with higher complementarity values. In addition, the functional annotations ratio of Gram+ were found significant lower than the Gram- group indicating their under-representation in the current databases.

We used feature selection to identify the pathways for which the relationship between the interaction indices and the growth yield was most prominent. To improve the selection we applied three distinct models, and summarised the results in an ensemble. The resulting selection of pathways was visually inspected using the pathway graph maps (e.g., Figure 4). This was possible because the size and complexity of the synthetic UTI community was moderate. We searched in these pathways for compounds, for which the KOs building up to compounds of interest were present in most groups but not in enterococci. In Table 2 we mention a few metabolites that comply with the above, and propose that they could be potentially complemented in enterococci by other UTI species.Transporters of those metabolites were found in the enterococci strains. For that task, we used the KEGG database to find the availability of the transporters, although more detailed annotations may be obtained from other databases or by employing specific tools ([13,55]). The hypothesised complementations may be identified experimentally, by assessing the presence of the compounds in the conditioned medium, or by measuring the growth of enterococci after addition of the compound to its medium. Alternatively, the hypothesis could be tested first on genome-scale metabolic models, such as the *E.*
*faecalis* V583 model [56]. For instance, this model suggest that enterococci have a histidine auxotrophy. Interestingly, the upregulation of sulphur, glutathione, and glycolytic metabolism (increasing metabolite precursors such as glycine and glutamate), and increased fatty acid synthesis, pathways that appear with high complementarity towards enterococci, are responses of *E.*
*faecalis* to oxygen [45]. It was reported that UTI found to cause oxidative stress [57].

A large part of the literature about microbial communities take growth rate as the target in their studies [58]. The relationship between growth rate and yield is still an open question [59], therefore it would be insightful to study how both growth characteristics are affected by the interaction between bacteria. Due to the nature of pairwise analysis, higher order interactions, i.e., those between more than two species, are not accounted for [60]. Although this is a limitation, pairwise studies may still be useful to understand the microbial activity and dynamics of more complex communities [61]. Furthermore, host-microbe interactions may have a large influence in the development of diseases such as UTIs and, therefore, be of interest for future exploration [61,62].

In this work, we studied the relation between genome-derived variables and the pairwise growth of bacteria isolated from UTI. We aimed to explore the extent to which the genomic information was enough to gain insight in pairwise interactions. To this end, new measures were created, based on competition and complementarity as mentioned in the literature. This allowed us to study correlations between those measures and pairwise growth effects. The complementarity measure with the largest correlation was further used to select pathways of relevance using feature selection. The analysis of these pathways made it possible to generate testable hypothesis, such as enterococci being complemented by other members of the UTI community, e.g., with folic acid. The supplementation of this compound led to the increase in the population density of UTI derived *E.*
*faecium* and *E.*
*faecalis* isolates. The verification of which UTI species are producing folic acid remains open, although our predictions alongside literature indicates potential candidates, such as *K.*
*pneumoniae* [63,64], *Pantoea* spp., and *Pseudomonas* spp. [65]. The complementarity of enterococci with folic acid from other species present in polymicrobial UTIs could also potentially explain the measured increased tolerance to the antibiotic trimethoprim-sulfamethoxazole [5]. The mode of action of this antibiotic combination is based on inhibiting DNA synthesis by limiting folate synthesis in bacteria [66]. Based on other studies it is known that enterocci are uniquely able to absorb folate from the environment, which gives them a fitness advantage in an trimethoprim-sulfamethoxazole environment when folate is present [67]. Yet, further research on the *in vitro* observed complementation of metabolic compounds, as well as other potential growth effects of community members in polymicrobial communities needs to be performed to assess the *in vivo* relevance. Moreover, further research is necessary to understand a large part of unexplored interactions that occur, especially the negative interactions. Finally, we believe that future steps to identify interactions between members in microbial communities will open many new avenues.

## Figures and Tables

**Figure 1 genes-12-01221-f001:**
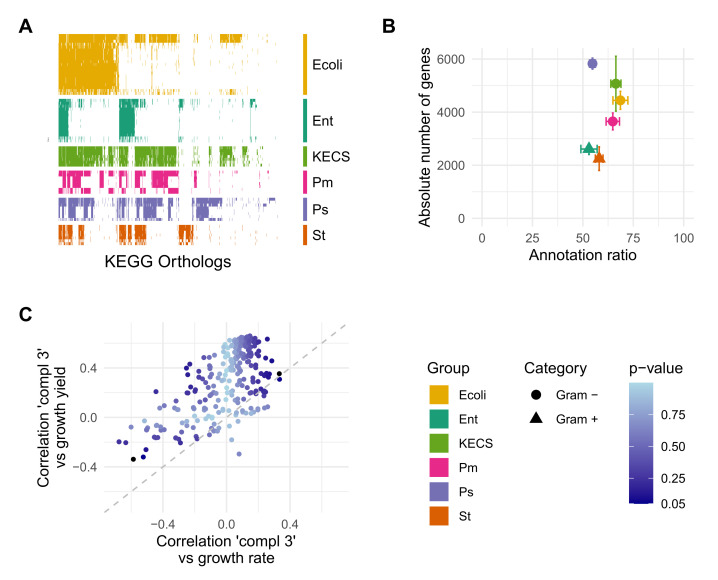
(**A**) Heatmap of presence (coloured) or absence (white) of KEGG Orthologs in the different bacteria groups. Although most KOs are present in the *KECS* group, other groups lack many. Note that only the KOs that are present in at least one organism are shown. (**B**) Although the absolute number of genes in the genome of each group varies per group, their annotation ratio from genome to KOs remains similar around 60%. The groups *St* and *Ent* have notoriously less genes and, thus, less absolute number of KOs than the rest of the groups. (**C**) Comparison of the relation of ‘Complementarity 3’ with the increase in two different growth measures: yield and rate (from [5]). The correlation is higher, for most pathways, when calculated against growth yield. An overview of the relation of all calculated indices with growth rate and yield is presented in Appendix A.

**Figure 2 genes-12-01221-f002:**
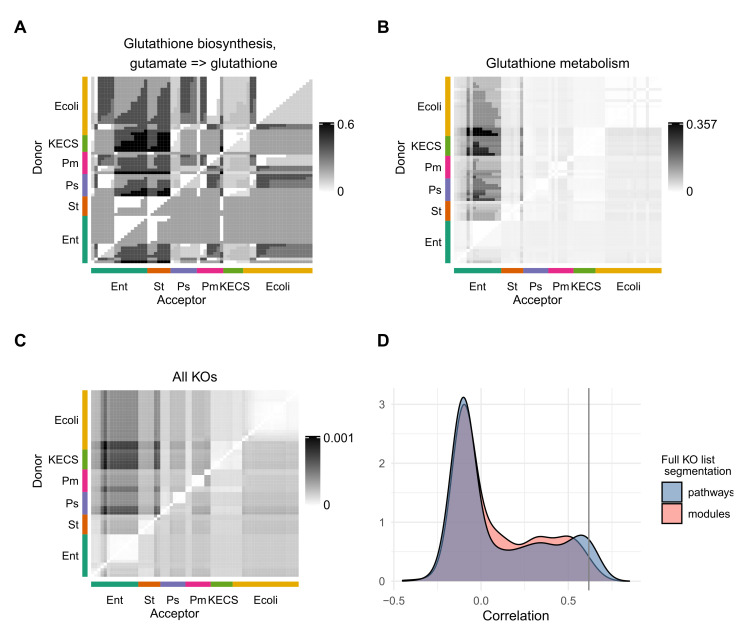
Heatmap of the ‘Correlation 3’ between the different strains within the groups. (**A**) The KOs selected belong to a module. (**B**) The KOs represented are part of a pathway. (**C**) All KOs were taken into account to calculate the ‘Complementarity 3’. (**D**) The density distribution of the correlations between the ‘Complementarity 3’ calculated using KOs belonging to one pathway at a time is shown in blue. In red, there is the similar calculation for modules. The vertical line represents the correlation using all available KOs. Using the pathways for the segmentation of the KO vectors yields the highest correlations.

**Figure 3 genes-12-01221-f003:**
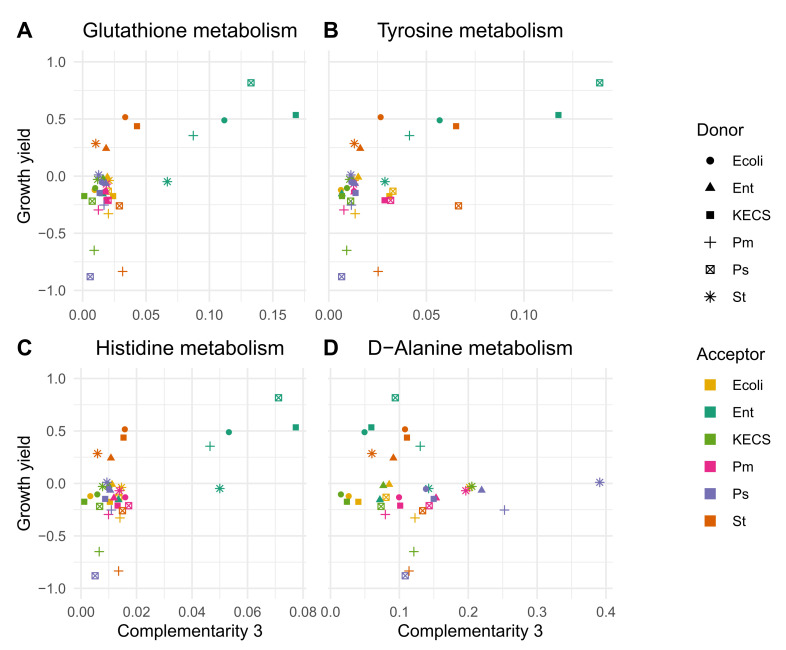
Relation between ‘Complementarity 3’ and growth yield (i.e., increase in the acceptor’s growth yield as seen in [5]). Each dot represents a pair acceptor-donor. This information is described in the right-hand legend. The first two pathways, (**A**) and (**B**), belong to the top-rank pathways in Table 1. They show a higher complementarity for the pairs for which the growth yield is higher, and vice versa. (**C**) represents the Histidine metabolism pathway. It was selected as another pathway of interest, and it shows a similar behaviour. On the other hand, in (**D**), the pathway D-Alanine metabolism was chosen as a low-rank pathway. In this case, there is no correlation between the complementarity measure and the growth yield increase.

**Figure 4 genes-12-01221-f004:**
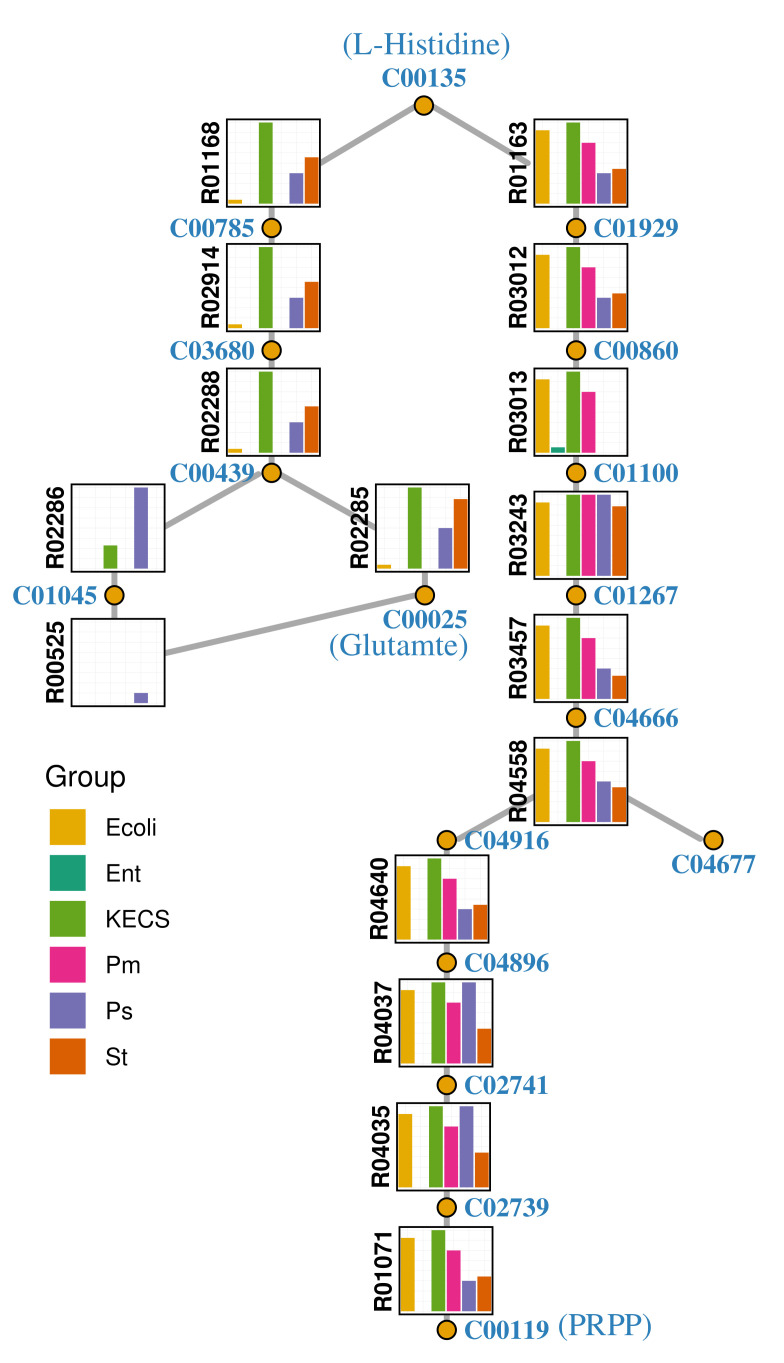
Network representation of histidine metabolism adapted from the corresponding KEGG histidine metabolism pathway (map00340). Circles correspond to compounds and bar plots to reactions. The percentage of species belong to each genera that possess the KEGG Ortholog associated with each reaction is represented by the size of the bar, coloured according to taxonomy inside the bar plot. In this example, coherently with the high complementarity value of enterococci, all the genera share a pathway towards biosynthesis of histidine, except for enterococci. C00119 is equivalent to 5-Phospho-alpha-D-ribose 1-diphosphate (PRPP). C04677 is equivalent to 5’-Phosphoribosyl-5-amino-4-imidazolecarboxamide (AICAR).

**Figure 5 genes-12-01221-f005:**
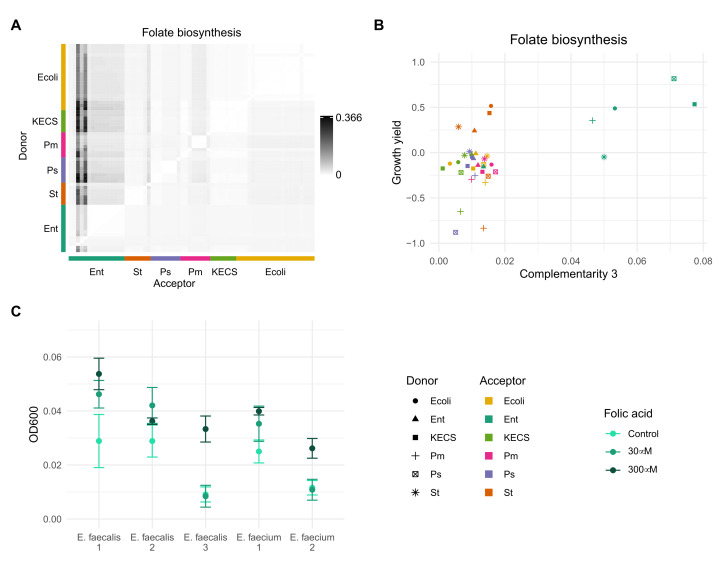
Folate complementarity. (**A**) Heatmap of Complementarity 3 for the pathway. (**B**) Enterococci as acceptors have higher complementarity for folate biosynthesis and increased growth yield values. (**C**) Supplementation of artificial urine medium with folic acid increases the growth yield of different *E. faecalis* and *E. faecium* isolates. Statistical significance was assessed with a *t*-test (*p* < 0.05), with Bonferroni correction to correct for multiple comparisons. All isolates, except *E. faecalis* 1, had a significant growth increase for at least one assessed folic acid concentration; *E. faecalis* 2 (30 μM), *E. faecalis* 3 (300 μM), *E. faecium* 1 (30 μM), *E. faecium* 2 (300 μM).

**Table 1 genes-12-01221-t001:** Pathways with high complementarity for the pairs with increased growth yield. Top ranking pathways according to the RF feature importance, the SVM coefficients, the MWU *p*-values, and the ensemble of the previous ranks.

Pathways	RF	SVM	MWU	Ensemble				
	**Feat. Imp.**	**Rank**	**Coef.**	**Rank**	***p*** **-Values**	**Rank**	**Sum**	**Rank**
Lipopolysaccharide biosynthesis	0.111	1	2.265	1	0.0003	13	15	1
Tyrosine metabolism	0.028	11	0.531	6	3.08 × $10^{-5}$	1	18	2
Sulfur metabolism	0.058	6	0.888	3	0.0003	12	21	3
Cationic antimicrobial peptide (CAMP) resistance	0.036	9	0.513	9	0.0001	6	24	4
Phenylalanine metabolism	0.073	4	0.313	16	0.0002	10	30	5
Biotin metabolism	0.017	15	0.385	14	7.01 × $10^{-5}$	4	33	6
Glutathione metabolism	0.015	17	0.243	22	9.44 × $10^{-5}$	5	44	7
Bacterial secretion system	0.050	7	0.905	2	0.001	41	50	8
Biofilm formation-Pseudomonas aeruginosa	0.020	14	0.318	15	0.0007	29	58	9
Riboflavin metabolism	0.007	23	0.518	8	0.0007	28	59	10
Histidine metabolism	0.014	18	−0.107	65	0.0003	14	97	22
Folate biosynthesis	0.001	53	0.016	48	0.0003	15	116	31

**Table 2 genes-12-01221-t002:** Compounds proposed to be complemented in enterococci by other UTI species. The KEGG pathway images referenced in the text are (1) Alanine, aspartate, and glutamate metabolism; (2) Arginine biosynthesis; (3) Sulphur metabolism; (4) Histidine metabolism; (5) Glycine, serine, and threonine metabolism; (6) Folate biosynthesis; (7) One carbon pool by folate.

Compound	Biological Relevance
Fumarate	Two ways to produce fumarate: from L-Aspartate (1) and from citruline (2), are not present in enterococci and staphylococci, but are in the others.
Sulphide	Sulphide can be formed from extracellular sulfate by many forms. Ent and Ecoli do not have the necessary orthologs, but they are able to convert this sulphide in Acetate and L-Cysteine (3). Cysteine is an amino acid, which has antioxidant activity [45].
Histidine, Glutamate, and Serine	Not directly synthesised in Enterococcus (4, 4, 5, respectively).
Folate (DHF) and derivates (10 formyl THF)	Routes available for production of folate (DHF) and derivates (10formylTHF) were missing in Enterococcus but not in the other genera. (6 (folate) and 7 (derivates)). Folate is an essential cofactor for nucleotide and amino acid biosynthesis. This makes folate a candidate for the hypothesis of complementarity. In addition, in mammalian cells there are folate transporters, however, in most bacteria those transporters do not exist and they need to synthesise their own folate, which make the biosynthesis of folate a target for antibiotics. However, some strains of enterococci are able to uptake exogenous folate [46,47], which would exclude them for the antibiotic target.

## Data Availability

The data and code used in the study can be found in GitHub (https://github.com/egarcialara/UTI_bacteria_interactions, accessed on 1 July 2021).

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
