# Peer review of "Using Functional Annotations to Study Pairwise Interactions in Urinary Tract Infection Communities"

_genes, 2021, doi:10.3390/genes12081221_

Round 1
Reviewer 1 Report
The currently reviewed manuscript by Lara et al. uses a computational approach to calculate pairwise interactions in urinary infection communities based on interaction indices derived by the authors. I acknowledge the great efforts of the authors to contribute to an understudied field (especially at the functional/genomic level). However, I have some serious reservations about the conclusions which have been presented. This work certainly is important to have in the scientific community. However, I believe some important points need to be addressed. I want to note that this work would be of great importance to the UTI/urinary microbiome field if sufficiently robust.
Major concerns:
- If I understand correctly, the current work relies on the assumption that maximizing the KO content in the pairwise association (particularly the number of unique pathways in a donor) is correlated with maximizing the growth potential of the accepting member of the relationship. I don’t necessarily believe this is an incorrect assumption. However, metabolic compensations for deficiencies are highly context dependent. I would encourage the authors to include some discussion about this assumption if it has been made. However, if I am incorrect in my assessment of this assumption, please feel free to rebut and emphasize something I perhaps misunderstood.
- Variance in genetic potential between members of the same genus can be massive (E. coli for example). There are at least 40 urinary or genitourinary E. faecalis strains on NCBI. There are also a number of complete UPEC (See ASM Microbiology Resource Announcements Belle M. Sharon et al. 2020) and uropathogenic Klebsiella strains on NCBI. Please comment on the rationale of using mostly non-urinary genomes. What if there are urinary tract-specific genomic enrichments among the members of urinary microbiota? How would this specifically affect your model and predictions? I feel that it is important to directly answer these questions. I understand that it is possible that these genomes were obtained in the past. However, unless further rationale can be supplied, I do not believe focusing the analysis at the genus level is sufficient to overcome this as the possibility of false positive might be high. How can you insure that the predictions this model makes are not spurious or relevant to the urinary tract? This issue is compounded further given that currently E. coli and E. faecalis have open pangenomes. Further detailed rationale as to why these genomes were chosen is needed to interpret the data presented here. I would also encourage the authors to incorporate more recently available urinary isolate complete genome data from NCBI.
- I do not find figure 5 C to be particularly interpretable as it lacks some transparency in what is being presented. What is the replicate n-value? Are these results reproducible? What is the variance? I would encourage the authors to perform an ANOVA (or non-parametric equivalent) with multiple comparisons post hoc to assess the statistical significance of this experiment. Line 351 uses the word “validate”. I think it is really good that the authors want to validate the predictions of the model with experimental data. However, I think this needs some work and transparency. I have 3 major issues with this validation attempt. 1) While we do observe E. faecium as a member of the urinary microbiome, it is rather uncommon compared to E. faecalis. Why was this particular single strain of E. faecium chosen as a representative for the genus Enterococcus? 2). The use of a single strain to validate the predictions of this work lacks robustness. At least a second strain (preferably a third) is needed to robustly validate these findings in vitro. The authors should draw a hard line between in vivo and in vitro validations. They are not necessarily translatable although in vitro evidence can be a strong suggestion of biological relevance in vivo (which is ultimately what is important). To be perfectly clear, I am not asking the authors to develop and use and animal model. I would suggest bulking the robustness of this validation with a second (preferably a third or more) Enterococcus isolate (perhaps E. faecalis). 3). In figure 5C, the data is presented as ‘Change in population size’. This data looks like fold change and the text uses “2.5 fold”. Is this fold change relative to control? I would suggest showing the actual OD600s here or include them in the supplement. Biological relevance is hard to interpret as is presented in this instance.
Minor concerns:
- Line 318 segmentation is misspelled. I would suggest an additions spell check of the manuscript. I did not thoroughly check the text for grammar and spelling errors this one just suck out.
Author Response
The currently reviewed manuscript by Lara et al. uses a computational approach to calculate pairwise interactions in urinary infection communities based on interaction indices derived by the authors. I acknowledge the great efforts of the authors to contribute to an understudied field (especially at the functional/genomic level). However, I have some serious reservations about the conclusions which have been presented. This work certainly is important to have in the scientific community. However, I believe some important points need to be addressed. I want to note that this work would be of great importance to the UTI/urinary microbiome field if sufficiently robust.
Major concerns:
- If I understand correctly, the current work relies on the assumption that maximizing the KO content in the pairwise association (particularly the number of unique pathways in a donor) is correlated with maximizing the growth potential of the accepting member of the relationship. I don’t necessarily believe this is an incorrect assumption. However, metabolic compensations for deficiencies are highly context dependent. I would encourage the authors to include some discussion about this assumption if it has been made. However, if I am incorrect in my assessment of this assumption, please feel free to rebut and emphasize something I perhaps misunderstood.
The reviewer is correct about our assumption. We added the following discussion at L443:
“On one hand, we assumed the complementarity indices maximize the difference of KO content in the pairwise association and therefore the number of exchangeable compounds. As such, the community dependency from the environment is minimized, reflecting cooperation and likely also robustness. Cooperative, cross-feeding interactions are especially likely to play a role in poor resource environments, such as urine (or AUM). On the other hand, we assumed competition indices minimize the difference of KO content in the pairwise association and therefore display the similarity of required compounds for all members of a given community (Machado et. al. 2021).“
Machado et. al. showed an environment-dependent differentiation between competitive and cooperative metabolism of the co-occurring members in microbial communities across thousands of habitats.
- Variance in genetic potential between members of the same genus can be massive (E. coli for example). There are at least 40 urinary or genitourinary E. faecalis strains on NCBI. There are also a number of complete UPEC (See ASM Microbiology Resource Announcements Belle M. Sharon et al. 2020) and uropathogenic Klebsiella strains on NCBI. Please comment on the rationale of using mostly non-urinary genomes. What if there are urinary tract-specific genomic enrichments among the members of urinary microbiota? How would this specifically affect your model and predictions? I feel that it is important to directly answer these questions. I understand that it is possible that these genomes were obtained in the past.
However, unless further rationale can be supplied, I do not believe focusing the analysis at the genus level is sufficient to overcome this as the possibility of false positive might be high. How can you insure that the predictions this model makes are not spurious or relevant to the urinary tract? This issue is compounded further given that currently E. coli and E. faecalis have open pangenomes.
The ideal scenario is to have the exact strains from de Vos et. al. 2017 sequenced and used with our method. As such, these particular genomes could be associated with the pairwise growth data or other metadata of interest. Instead, due to the unavailability of the genomes, we aimed to obtain a general idea of interactions possible between the genus of interest, not the exact and only interactions occurring between specific strains. This decision arose from the high variance in genetic potential between members of the same genus as the reviewer correctly pointed out. We add that such variance is high also between strain on species level.
We raised this point before and updated the text at L399: “Since the genome sequences of the UTI isolates observed in de Vos et. al. 2017 were unavailable, we used publicly available genomes of similar organisms. Therefore, we have to accept an increased bias towards strain-specific interactions.”
We further update the following discussion at L402:
“To compensate for this potential bias, we focused our analyses on the interactions conserved between genera. Furthermore, interactions outside of the core metabolism and potentially encoded by the pan-genome, like toxin synthesis or quorum sensing, are more likely to be missed since these are known to vary more between strains.”
Therefore, this was the reason that we wanted to “validate” our predictions. We provide complete code in the form of a Github to facilitate a strain-specific analysis when the genomes would be available, as well as to be used by other scientists on other studies.
Further detailed rationale as to why these genomes were chosen is needed to interpret the data presented here. I would also encourage the authors to incorporate more recently available urinary isolate complete genome data from NCBI.
In 2.2. Description of the dataset section, we explain our rationale on the genome selection. As the reviewer indicates, we as well aimed to have urinary tract-specific genomes.
L112: “When a species presented more strains available than the number of species present in the experimental community, the strains related with UTI according to literature were preferred. Otherwise, human isolates were chosen and, as a last resort, isolates from other sources were selected.”
Our selection at that time led to a smaller number of UTI related isolates. Therefore we followed the reviewer's suggestion and updated our data. We agree with the reviewer that such action may reduce the false positive, although it will not solve the problem of not having the exact genomes.
Accordingly, we replaced the strains that were not urine or human isolates. In total, there were 20 strains replaced: 7 E. coli, 7 enterococci, 3 staphylococci, 1 Pseudomonas aeruginosa and 2 from the KECS group. From the new genomes, 16 were from urine or UTI isolates, and the remaining 4 were human isolates. The complete selection was added in the appendix (Table S2).
After the rerun of the analysis on the updated dataset, we obtained highly comparable results with the original analysis.
We added the following text at L116:
“Table S1 describes the initial selection of species and the equivalent strains, of which the results are described below. Table S2 describes a selection of the same species, but this collection only contains urine or human derived isolates. The results of the pipeline on those two genome selections (described in Table S1 and S2) were very similar (Figure S7, Table S7).”
As we mentioned, the predictions of interaction are genus level specific and although we tested two datasets with the second composed of genomes closer to the UTI strains we did not find a strong environmental interaction signal.
We added the following text at L409: “This allowed us to use the genomes of strains not strictly associated with UTI to generate hypotheses on interactions between members of UTI communities. We obtained similar results compared to genomes more strictly related to UTI, which can be explained by the methods focus to identify core-genome interactions or indicate that the environmental interactions signal is weak”
- I do not find figure 5 C to be particularly interpretable as it lacks some transparency in what is being presented. What is the replicate n-value? Are these results reproducible? What is the variance? I would encourage the authors to perform an ANOVA (or non-parametric equivalent) with multiple comparisons post hoc to assess the statistical significance of this experiment.
We repeated the experiment, with multiple Enterococcus strains, specifically three Enterococcus faecalis, two Enterococcus faecium isolates. The growth complementation, the increase in population size upon supplementation of the artificial urine medium with folic acid, are reproducible, also for other strains, except for one (Enterococcus faecalis1). All these strains are derived from polymicrobial UTIs.
Previous experiments (we have omitted the previous experiments in the current version, and updated the data with the new experiments), and these experiments were all performed in triplicate. We added Bonferroni-correction to allow for the testing of multiple comparisons.
Line 351 uses the word “validate”. I think it is really good that the authors want to validate the predictions of the model with experimental data.
However, I think this needs some work and transparency. I have 3 major issues with this validation attempt. 1) While we do observe E. faecium as a member of the urinary microbiome, it is rather uncommon compared to E. faecalis. Why was this particular single strain of E. faecium chosen as a representative for the genus Enterococcus?
Thanks for pointing this out. The reason we used this particular Enterococcus faecium was simply due to the fact that we use it most for all other experiments in the lab, and that it showed a relatively strong effect on the conditioned medium from other UTI isolates before.
2). The use of a single strain to validate the predictions of this work lacks robustness. At least a second strain (preferably a third) is needed to robustly validate these findings in vitro. The authors should draw a hard line between in vivo and in vitro validations. They are not necessarily translatable although in vitro evidence can be a strong suggestion of biological relevance in vivo (which is ultimately what is important).
We agree with the point stated by the reviewer. Validating predictions with only one strain, while the predictions coming from an ensemble of strains is not sufficiently robust. And importantly, we fully agree that in vitro validations are no proof of in vivo effects. We, therefore, have added to the discussion:
L529: ‘Yet, further research on the in vitro observed complementation of metabolic compounds, as well as other potential growth effects of community members in polymicrobial communities needs to be performed to assess the in vivo relevance.’
To be perfectly clear, I am not asking the authors to develop and use and animal model. I would suggest bulking the robustness of this validation with a second (preferably a third or more) Enterococcus isolate (perhaps E. faecalis).
We thank the reviewer for explicitly dictating this experimental suggestion.
Therefore, we repeated the supplementation assays with four more isolates, three Enterococcus faecalis isolates, and one more Enterococcus faecium isolate. We found that supplementation of folic acid to the artificial urine medium did increase the population size of four out of five isolates (two Enterococcus faecalis isolates, two Enterococcus faecium isolates). For your reference Enterococcus faecium1 is the strain used in the initially submitted version of this manuscript.
3). In figure 5C, the data is presented as ‘Change in population size’. This data looks like fold change and the text uses “2.5 fold”. Is this fold change relative to control? I would suggest showing the actual OD600s here or include them in the supplement. Biological relevance is hard to interpret as is presented in this instance.
Indeed, fold change was a relative change in population size in the presence of folic acid compared to the control (in the absence of folic acid). For transparency, we replaced the fold-change data showing all data points using OD600 values. As you can see, the OD600 values for enterococci in artificial urine medium are rather low, as is the case in ‘real’ urine. In control artificial urine medium, enterococci are present at a concentration in the order of ~5.10^5 CFU/ml.
Minor concerns:
- Line 318 segmentation is misspelled. I would suggest an additions spell check of the manuscript. I did not thoroughly check the text for grammar and spelling errors this one just suck out.
We corrected the misspelled word and went through the text for further identification.
Reviewer 2 Report
In the present study, the authors predict key bacterial interactions in the urinary tract infection based on the genetic potential of the bacteria. However, to what extent these bacterial species truly interact and biosynthesize molecules within the urinary tract is questionable.
Is there any experimental evidence from previous studies about these microbe-microbe interactions and metabolic exchanges that authors are alluding to.
Is the metabolic reconstruction model that authors implemented also integrates metabolomics data alongside genetic potential of the microbes?
Any evidence that the microbes you selected co-occur in the urinary tract? in other words, positively correlate in their relative abundances in the gut. Any idea such evidence has been shown through human microbiome project studies via 16S/shotgun sequencing data? OR Is there any physical co-occurrence evidence or spatial organization of the microbes in the urinary tract.
Could authors indicate what might be the translation potential of the current findings.
Author Response
In the present study, the authors predict key bacterial interactions in the urinary tract infection based on the genetic potential of the bacteria. However, to what extent these bacterial species truly interact and biosynthesize molecules within the urinary tract is questionable.
Is there any experimental evidence from previous studies about these microbe-microbe interactions
The microbe-microbe interactions were tested in vitro before, as published at de Vos et. al. 2017. We use those experimentally derived data to guide our in silico predictions.
and metabolic exchanges that authors are alluding to.
Metabolic exchanges were not tested. The purpose of this computational method is to identify metabolites of interest that could be exchanged between the members of the community.
Is the metabolic reconstruction model that authors implemented also integrates metabolomics data alongside genetic potential of the microbes?
No, we do not construct metabolic models - this is for a future structure. In this work, we propose an initial step to identify putative interaction which could also be used to select the most appropriate strains to construct their genome-scale metabolic models.
Any evidence that the microbes you selected co-occur in the urinary tract? in other words, positively correlate in their relative abundances in the gut. Any idea such evidence has been shown through human microbiome project studies via 16S/shotgun sequencing data? OR Is there any physical co-occurrence evidence or spatial organization of the microbes in the urinary tract.
The species do co-occur, these species were isolated together in polymicrobial UTI communities by Croxall et. al. 2011, and their interactions were assessed in de Vos et. al. 2017.
To clarify this we added L73 "Communities containing four different species were isolated from hosts diagnosed with a polymicrobial UTI"
Could authors indicate what might be the translation potential of the current findings.
We state in L24: “Moreover, interactions between the members of a microbial community can potentially alter the sensitivity to antibiotics in vitro, and therefore potentially the effect of treatments (de Vos et. al. 2017).”
We believe that if this is also the case, then smart combinations of bacteria can increase the efficacy of antibiotic treatments, or recreate potential efficacy issues.
Round 2
Reviewer 2 Report
Interesting work